*Cambridge Prisms: Global
Mental Health*

# A bibliometric analysis of scientific literature on adverse childhood experiences (2004–2024)

B.K. Sai Sindhura [ID] and Ganesh Kumar J

Department of Psychology, Christ University, Bangalore, India

## Research Article

A bibliometric analysis of scientific literature on
adverse childhood experiences (2004–2024).
*Cambridge Prisms: Global Mental Health*, **12**,
e60, 1–11

ACE; bibliometrics; childhood adversity; global
production; knowledge mapping

**Corresponding author:**
B.K. Sai Sindhura;
Email: sai.bk@res.christuniversity.in

## Abstract

Adverse childhood experiences (ACE) significantly impact physical, mental and social well-being, making them a critical area of research. This study analyzed the emerging trends and intellectual structure of ACE research and identified key contributors, including the most productive nations, journals and authors. Using bibliometric tools and VOSviewer software (version 1.6.20), 1,957 articles from the Scopus database (2004 to March 2024) were systematically analyzed. A notable finding was the surge in ACE-related publications during the COVID-19 pandemic, potentially reflecting increased global attention on childhood adversity amid heightened social and economic challenges. The analysis also revealed a striking dearth of studies from the Global South, with the field predominantly shaped by Western nations, like the United States, the United Kingdom, Australia and Canada. Leading journals, such as the *Journal of Interpersonal Violence*, and prolific authors, like Kevin T. Wolff, played a central role in advancing the field. Co-citation analysis uncovered four thematic clusters: (1) conceptualization and assessment of ACE, (2) health implications, (3) mental health impacts and (4) juvenile delinquency. These clusters, though distinct, showed significant thematic overlaps, reflecting the interconnected nature of ACE research and its intellectual structure. These findings underscore the need for more regionally diverse and interdisciplinary approaches to understanding global childhood adversity.

## Impact statement

This study offers a critical overview of how global research on adverse childhood experiences (ACE) has evolved. By mapping influential nations, authors, journals and themes within ACE literature over the past two decades, this article identifies major knowledge clusters, regional disparities and thematic shifts. One of the key but unexpected contributions of this article is its demonstration of a sharp increase in ACE-related publications during the pandemic years. The findings from this study have important implications for both research and policy. For researchers, this article serves as a roadmap to understand where the field of ACE research has been and where it is going. For practitioners and policymakers, it provides evidence that supports the expansion of child protection systems and mental health services, particularly in times of social upheaval like the COVID-19 pandemic. Overall, this study bridges academic knowledge with real-world relevance, advocating for sustained, globally diverse and multidisciplinary efforts to reduce childhood adversity and its lifelong consequences.

## Introduction

Adverse childhood experiences (ACE) are potentially traumatic events that occur in a child's life before the age of 18 and include physical, emotional and sexual abuse, physical and emotional neglect, and household dysfunction, such as domestic violence, parental separation or divorce, witnessing domestic violence, substance abuse, incarceration of a family member (Felitti et al., 1998), as well as exposure to crimes or violence in the community (Lee et al., 2017). Dong et al.'s (2004) study provided strong evidence that the presence of one ACE significantly increased the prevalence of additional ACE and that these factors often interact and can have compounding effects on an individual's life.

Chronic and frequent exposure to ACE can have a profound and lasting impact on individuals' health and well-being, and is linked to the risk of chronic physical diseases, such as heart disease, diabetes, cancer and obesity (Holman et al., 2016; Soares et al., 2021), as well as to mental health challenges, such as depression, anxiety and PTSD, as well as feelings of helplessness, lower resilience and fear (Oral et al., 2016). Furthermore, individuals who have experienced ACE are more likely to engage in risky behaviors and unhealthy coping mechanisms, such as substance abuse, early sexual activity and delinquency (Boullier and Blair, 2018), while also negatively impacting cognitive function and academic achievement (Metzler et al., 2017). This makes the

concept of ACEs is an essential framework for understanding how and why an individual's life turned out the way it did; it explains the plausible reasons for the various physical, emotional, social or behavioral issues that a person may be facing as an adult (Dube et al., 2003; Wolff K and Baglivio, 2017; Hails et al., 2022).

The study of ACE has a rich history in psychology, rooted in the seminal research conducted by the Centers for Disease Control and Prevention (CDC) and Kaiser Permanente in 1998. This landmark study by Felitti et al. defined ACE across 10 categories, including abuse (physical, emotional and sexual), neglect (physical and emotional) and household dysfunction (domestic violence, parental separation or divorce, substance abuse, incarceration and mental illness). It demonstrated a strong, graded relationship between the number of ACE experienced and a range of adverse health and social outcomes in adulthood, laying the foundation for subsequent research. Building upon this foundational work, the Philadelphia Expanded ACE Study by Cronholm et al. (2015) reconceptualized ACE by incorporating additional community-level stressors, such as neighborhood violence, racism, living in foster care and exposure to unsafe environments. This broader perspective highlighted the critical role of social and environmental factors in shaping childhood adversity, emphasizing the potential for intergenerational trauma and the complex interplay between individual and systemic influences on well-being.

In terms of publication trends, Struck et al. (2021) conducted the first bibliometric analysis of ACE literature, analyzing 789 articles published in 351 journals between 1998 and 2018. Their study revealed a significant increase in ACE-related publications over two decades, with general medicine and multidisciplinary research emerging as the most prominent fields. It also identified mental and physical health as the primary outcomes of interest, reflecting the health-focused lens through which ACE have predominantly been studied. However, while these contributions have shaped the field, there remains a need for more inclusive bibliometric studies that extend the scope of analysis to include recent trends, emerging themes and global contributions.

### Rationale and objectives

While Struck et al. (2021) provided an important foundation by analyzing publication trends in ACE research from 1998 to 2018, it focused predominantly on the volume of research, primary outcomes and populations studied, without delving into the intellectual structure or identifying key contributors, such as influential nations, journals and authors driving the field. Similarly, systematic reviews (e.g., Kalmakis and Chandler, 2015; Hughes et al., 2017) have synthesized existing knowledge on ACE and their health consequences but are inherently limited in their capacity to reveal the broader intellectual landscape or the dynamics of research growth, collaboration and influence in the domain.

The current review addresses these gaps by extending the analysis to a broader time frame (2004 to March 2024), thereby capturing the most recent trends and emerging themes in ACE research. This study's first objective was to identify emerging themes, the most productive and influential authors, journals and nations, as it sheds light on those who have made significant contributions to the development of ACE research. Productivity, typically measured by the number of articles published, indicates sustained engagement and contribution to the academic discourse. While productivity does not automatically imply quality or impact, it serves as an important indicator of an active role in advancing the field. Equally important is the role of citations, which are a widely accepted metric for assessing the impact or influence of scholarly

work. The number of citations a publication receives reflects its significance and the extent to which it has shaped subsequent research. In the context of ACE research, citation counts serve as an indicator of how much influence a specific author, paper or journal has had within the academic community. By incorporating both productivity and citations, this study maps the key contributors – authors, journals and nations – in ACE research.

The second objective was to study the intellectual structure of ACE research, which contributed to understanding the dominant research themes, and the relationships between them, and highlighting areas of potential collaboration, growth or underexplored opportunities. These findings may offer valuable insights to researchers, mental health professionals and policymakers interested in ACE.

## Methods

Bibliometric analysis is a quantitative research method that examines patterns in academic publishing, such as publication output and citation trends, to evaluate the structure and development of a field (Broadus, 1987). According to Donthu et al. (2021), this approach is a useful tool for handling large datasets, enabling a thorough understanding of a field, identifying gaps in knowledge and strategically positioning scholarly contributions. Bibliometric analysis has been increasingly applied across diverse fields of science to map research landscapes and identify emerging trends. For instance, studies have explored research patterns in areas such as One Health (Miao et al., 2022), climate change (Suhaimi and Mahmud, 2022) and the Sustainable Development Goals (Mishra et al., 2024). These examples underscore the method's adaptability in capturing complex, multidisciplinary themes and informing evidence-based decision-making across sectors.

For the purposes of this study, specific bibliometric tools, such as descriptive and citation analyses, were utilized to achieve the first research objective, while co-citation analysis of articles was employed to accomplish the second objective. The present study did not include any human subjects; hence, ethical approval was not required on the basis of the guidance received from the institutional review board of Christ University, India.

### Keyword search

The search strategy for this bibliometric analysis focused on identifying academic articles that investigated ACEs and their associated impacts. To ensure comprehensive coverage of the relevant literature, the primary search keywords used in this analysis were "adverse childhood experiences" OR "ACE". The rationale for selecting these keywords is twofold. First, "adverse childhood experiences" is the most commonly used term in the literature to describe the broad category of childhood traumas and stressors that have been linked to long-term health and developmental consequences. Second, "ACE" is the commonly accepted abbreviation of "adverse childhood experiences" and is frequently used in academic publications. Using both the full term and the abbreviation ensured that the search captured articles employing either phrase.

### Database selection

Bibliometric analyses usually rely on sourcing relevant data from multiple databases. However, we decided to utilize Scopus as the sole database for this bibliometric analysis due to its comprehensive

coverage, high-quality indexing and relevance to the research objectives. Moreover, Donthu et al. (2021) emphasize selecting a single appropriate database for bibliometric analysis as it eliminates the need for consolidating data from multiple sources. This approach reduces the risk of duplication, inconsistencies and potential human errors during data processing. Given that Scopus is widely recognized for its robust metadata and advanced bibliometric tools, it was deemed the most suitable database for ensuring trustworthiness and consistency in the analysis.

### Data extraction and cleanup

To ensure comprehensive data coverage, we conducted the search across all available fields – specifically titles, abstracts and keywords – using the keywords described above, between 2004 and March 2024. The rationale for the time frame chosen for this bibliometric analysis was grounded in the evolution of ACE research and its growing prominence over the past two decades. Although ACE were first introduced by Felitti et al. (1998), their conceptualization gained significant traction in academic and policy discussions starting in the early 2000s. This period marked the establishment of ACE as a critical area of research, with a surge in studies exploring their public health implications, intergenerational effects and societal costs.

The inclusion criteria limited the data considered to those in English in refereed journals and listed in the "Psychology," "Social Science" and "Health Professions" subject areas only. We made the decision to focus on research primarily within social sciences, psychology and health, intentionally excluding studies from certain disciplines, such as medicine, neuroscience, nursing and other related fields. This exclusion was based on the desire to focus on the broader, non-clinical aspects of ACE research, particularly those relating to societal, psychological and intergenerational factors, which are often less emphasized in clinical or medical-focused research. Additionally, we sought to identify intellectual trends specific to fields such as social science, and public health, which may not always overlap with medical or neuroscience-focused research.

To confirm the integrity of the bibliographic references, we confirmed the existence of each entry in the exported data. Discrepancies such as book reviews and dissertations that were incorrectly categorized as journal articles were resolved. Additionally, an assessment of titles and abstracts led to the exclusion of data that were completely unrelated to the field. After cleanup, 1,957 entries were selected for the current analysis from the original raw file, as detailed in Figure 1.

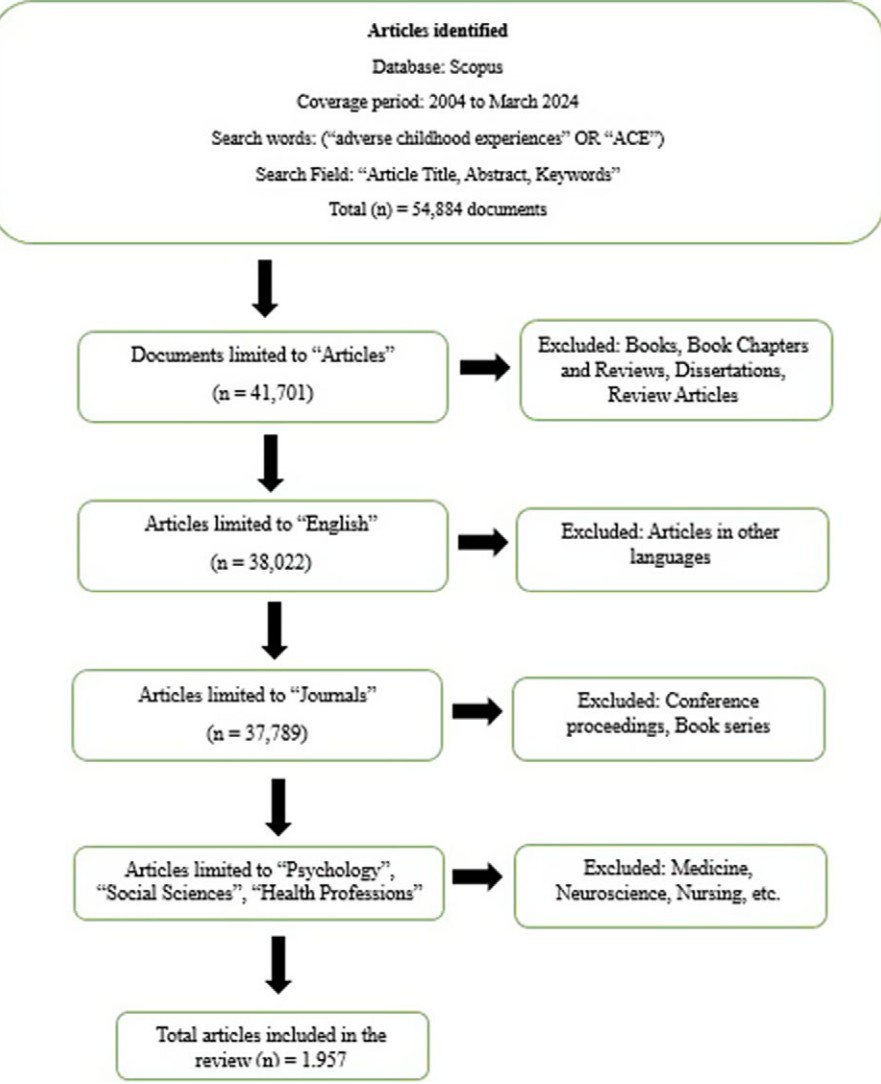

**Figure 1.** Flowchart of the article selection process.

### Data analysis

The analysis was carried out using two applications – Microsoft Excel 2019 and the VOSviewer software (version 1.6.20), a bibliometric analysis and visualization tool (Eck et al., 2010). Microsoft Excel was primarily used to clean up the data and conduct descriptive statistical analyses such as publication trends over the years, authorship and journal information. VOSviewer was used to generate visual maps of the intellectual structure of the field, through the co-citation of articles.

## Results

### Descriptive and citation analyses

The sample for this study consisted of 1,957 articles authored by 6,490 authors in 663 journals listed in the Scopus database between 2004 and March 2024. As shown in Figure 2, there has been a steady rise in the number of publications on ACE since 2004. However, the trend particularly showed an upward trajectory and saw a sudden spike in article production from 2020 to 2023, which coincides with the global COVID-19 pandemic.

Among the most productive countries, the United States (1,126) stood first, followed by the United Kingdom (160), Australia (114) and Canada (107). The United States (16,748) followed by the United Kingdom (2,404), Australia (1262) and Canada (1,177) were also the countries with the greatest number of citations and, hence, are the most impactful in this area of study. Table 1 shows the top 10 countries with the highest productivity and impact. Note that none of the African or South American nations are on the list.

Table 2 lists the top 10 journals that have published more than 10 articles on ACE and the most impactful journals with more than 200 citations. The *Journal of Interpersonal Violence* had the highest number of articles (100) related to ACEs, followed by *Frontiers in Psychology*, *Children and Youth Services Review* and *Psychological Trauma: Theory, Research, Practice and Policy* with 74, 73 and 62 articles, respectively. However, *Children and Youth Services Review* remained the most influential journal with 1,348 citations, followed by the *Journal of Interpersonal Violence* (1,211 citations),

**Table 1.** Most productive and impactful countries

| Most productive countries | | Most impactful countries | | |
|---|---|---|---|---|
| Country | No. of articles | Country | Citations | Avg. citations per article |
| United States | 1,126 | United States | 16,748 | 14.84 |
| United Kingdom | 160 | United Kingdom | 2,404 | 15.03 |
| Australia | 114 | Australia | 1,262 | 11.07 |
| Canada | 107 | Canada | 1,177 | 11.00 |
| China | 59 | South Africa | 717 | 21.73 |
| Germany | 39 | Germany | 536 | 13.74 |
| Spain | 35 | Italy | 402 | 12.56 |
| South Africa | 33 | France | 396 | 19.80 |
| Italy | 32 | Spain | 352 | 10.06 |
| South Korea | 27 | China | 341 | 5.78 |

*Youth Violence and Juvenile Justice* (883 citations), *Frontiers in Psychology* (796 citations), the *Journal of Youth and Adolescence* (647 citations), *Psychological Trauma: Theory, Research, Practice, and Policy* (631 citations), the *Journal of Criminal Justice* (624 citations) and the *Journal of Clinical Child and Adolescent Psychology* (419 citations).

Table 3 lists the most productive and impactful authors in this domain. With 17 articles, Kevin T. Wolff (John Jay Research and Evaluation Center, USA) was the most productive author, followed by Melissa S. Jones (Brigham Young University, USA), who has authored 15 articles. Hayley Pierce (Brigham Young University, USA) occupied the third position with 10 articles published, and Trevor Spratt (Trinity College Dublin, Ireland), with nine published articles, occupied the fourth position.

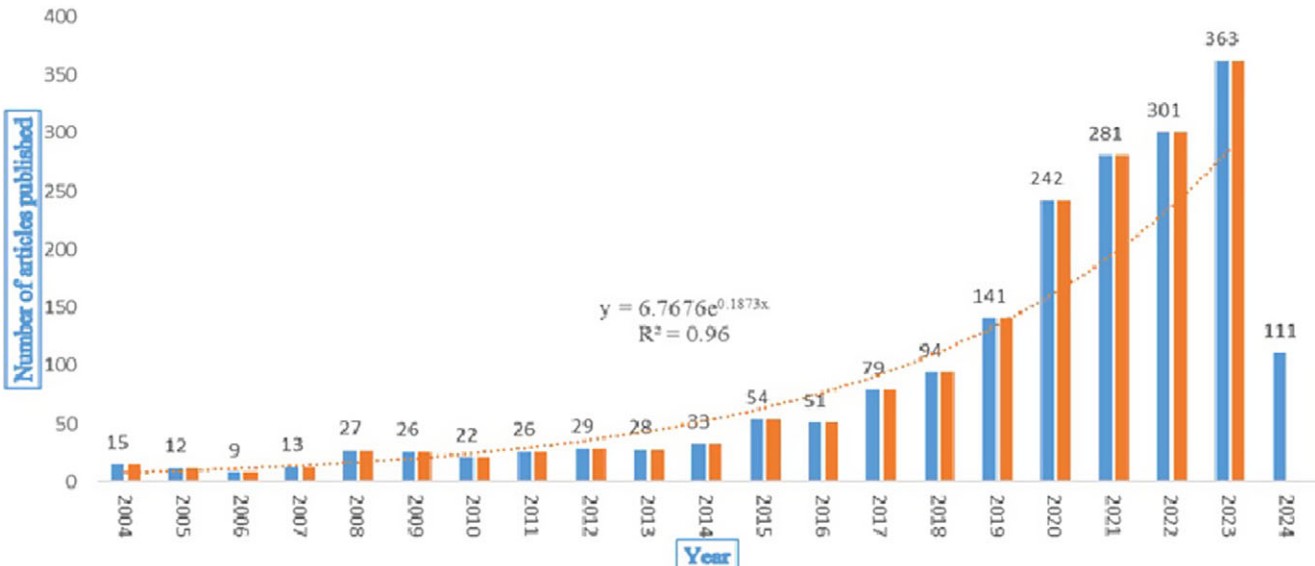

**Figure 2.** Article production from 2004 to March 2024.

**Table 2.** Most productive and impactful journals

| Most productive journals (>10 articles) | | Most impactful journals (Citations >200) | | |
|---|---|---|---|---|
| Journal (impact factor) | No. of articles | Journal (impact factor) | Citations | Avg. |
| *Journal of Interpersonal Violence* (2.6) | 100 | *Children and Youth Services Review* (2.4) | 1,348 | 18.47 |
| *Frontiers in Psychology* (3.8) | 74 | *Journal of Interpersonal Violence* (2.6) | 1,211 | 12.11 |
| *Children and Youth Services Review* (2.4) | 73 | *Youth Violence and Juvenile Justice* (1.5) | 883 | 31.54 |
| *Psychological Trauma: Theory, Research, Practice and Policy* (2.7) | 62 | *Frontiers in Psychology* (3.8) | 796 | 10.76 |
| *Youth Violence and Juvenile Justice* (1.5) | 28 | *Journal of Youth and Adolescence* (3.7) | 647 | 28.13 |
| *Current Psychology* (2.3) | 25 | *Psychological Trauma: Theory, Research, Practice and Policy* (2.7) | 631 | 10.18 |
| *Journal of Family Violence* (2.7) | 24 | *Journal of Criminal Justice* (3.3) | 624 | 31.20 |
| *Journal of Youth and Adolescence* (3.7) | 23 | *Journal of Clinical Child and Adolescent Psychology* (4.2) | 419 | 69.83 |
| *Journal of Criminal Justice* (3.3) | 20 | *American Psychologist* (12.3) | 267 | 26.70 |
| *British Journal of Social Work* (1.9) | 12 | *Journal of Family Violence* (2.7) | 233 | 9.71 |

*Note*: "Avg." indicates the average citations per article.

Kevin T. Wolff (John Jay Research and Evaluation Center, USA) was also the most influential author in the field, with 842 total citations and an average of 49.53 citations per article. This was

followed by Melissa T. Merrick (Prevent Child Abuse America and National Center for Injury Prevention and Control at the CDC, USA) with 634 citations and Katie Ports (American Institutes for Research, USA) with 440 citations. Paula Nurius (University of Michigan, USA) and Myriam Forster (California State University, USA) take the fourth and fifth positions for the most impactful authors on ACE, with 279 and 206 citations, respectively.

## Co-citation analysis of articles

The intellectual structure of ACE research was identified through co-citation analysis, a bibliometric technique that examines how frequently pairs of articles are cited together. Co-citation analysis is widely used for exploring the conceptual framework and intellectual structure of a research field (Marshakova-Shaikevich, 1973; Small, 1973). This technique identifies correlations between previously published articles by analyzing how frequently they are cited together in subsequent studies (Donthu et al., 2021). Such analysis is instrumental in uncovering clusters of interrelated research themes, offering deeper insights into the evolution, development and structure of the field.

Given the study's objective to "decipher the intellectual structure of ACE research," co-citation analysis was conducted on the dataset of 1,957 articles using the VOSviewer software. The analysis revealed four distinct clusters, representing major thematic areas in the field of ACE research – Cluster 1: Concept and Assessment of ACE; Cluster 2: ACE and Health Implications; Cluster 3: ACE and Mental Health; and Cluster 4: ACE and Society. These clusters collectively map the intellectual structure of ACE research, providing an in-depth understanding of the key themes that have shaped the field over time. Figure 3 presents the network diagram of the co-citation analysis, and Table 4 highlights the key articles within each cluster.

## Cluster 1: Concept and Assessment of ACE

Articles in this cluster dealt with the re-conceptualization and the assessment of ACE. The papers explored critical questions about clarifying the concept of ACE, the nature and scope of adversities during childhood and a critique of existing methods to measure ACE. By exploring the fundamental nature of ACE and

**Table 3.** Most productive and impactful authors

| Most productive authors (≥5 articles) | | | Most impactful authors (top 10) | | |
|---|---|---|---|---|---|
| Author | Affiliation | No. of articles | Author | Citations | Avg. |
| Kevin T. Wolff | John Jay Research and Evaluation Center, USA | 17 | Kevin T. Wolff | 842 | 49.53 |
| Melissa S. Jones | Brigham Young University, USA | 15 | Melissa T. Merrick | 634 | 211.33 |
| Hayley Pierce | Brigham Young University, USA | 10 | Katie Ports | 440 | 110.00 |
| Trevor Spratt | Trinity College Dublin, Ireland | 9 | Paula Nurius | 279 | 55.80 |
| Elizabeth Crouch | University of South Carolina, USA | 7 | Myriam Forster | 206 | 29.43 |
| Myriam Forster | California State University, USA | 7 | Michelle Kelly-Irving | 192 | 48.00 |
| Paula S. Nurius | University of Washington, USA | 5 | Melissa S. Jones | 164 | 10.93 |
| Elizabeth Radcliff | University of South Carolina, USA | 5 | Trevor Spratt | 153 | 17.00 |
| Canan Karatekin | University of Minnesota, USA | 5 | Barbara McMorris | 139 | 46.33 |
| John Devaney | University of Edinburgh, UK | 5 | Hayley Pierce | 121 | 12.10 |

*Note*: "Avg." indicates the average citations per article.

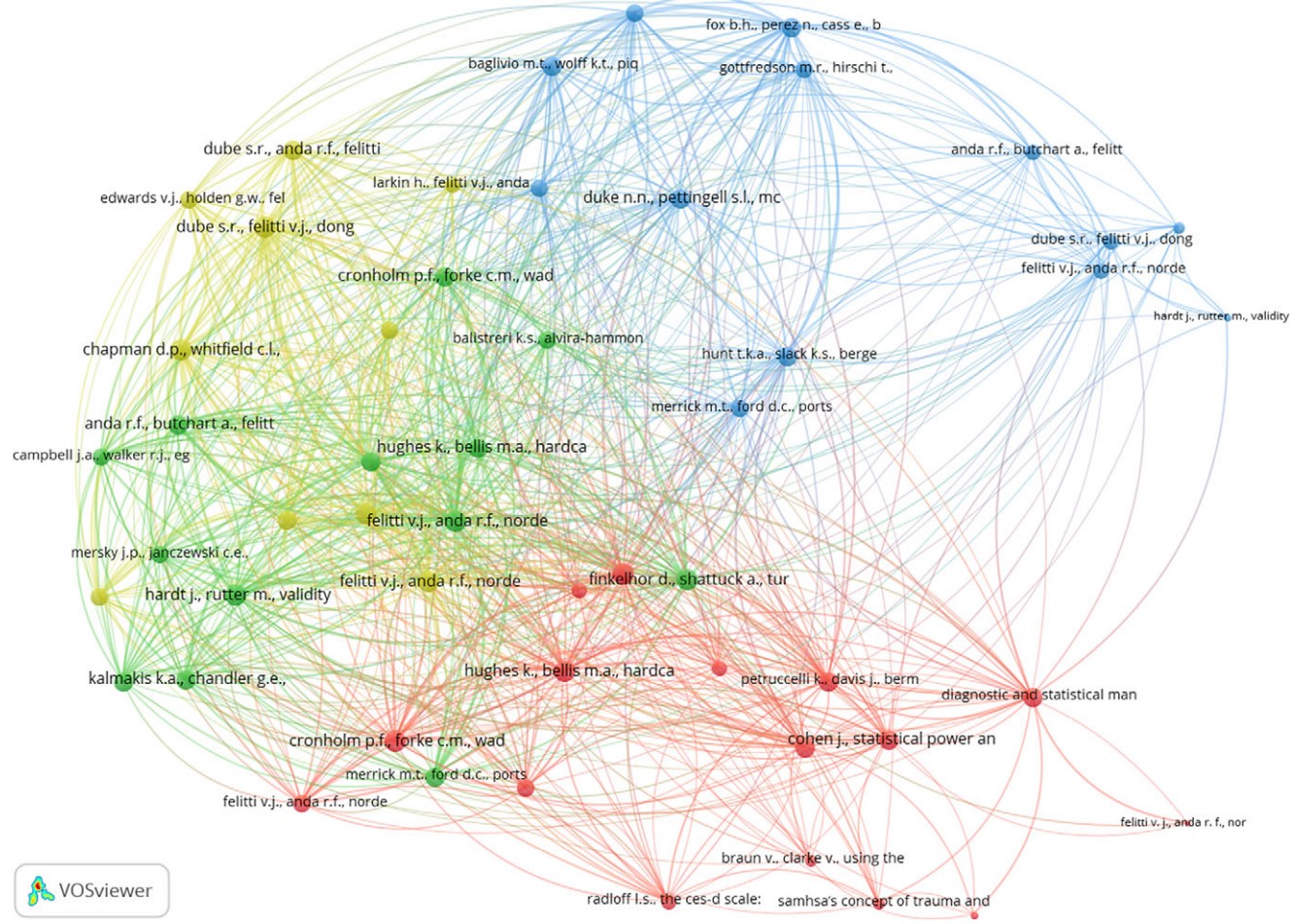

**Figure 3.** Article co-citation analysis map.

re-examining standardized tools for measurement, this cluster formed the foundation of research on ACE between 2004 and March 2024.

Cronholm et al. (2015) argued that knowledge about ACE predominantly relied on data collected from white middle-/upper-middle-class participants with a focus on experiences within the home. By employing a broader spectrum of participants from various socioeconomic and racial backgrounds, Conventional and Expanded (community-level) ACE were established as necessary to capture the extent of adversity adequately (Cronholm et al., 2015). This contributed to the expansion of the definition of ACE and the much-needed contextual reconceptualization. Similarly, Kalmakis and Chandler (2014) arrived at a new conceptual definition of ACE through a systematic review of the literature. They defined ACE as "childhood events, varying in severity and often chronic, occurring in a child's family or social environment that cause harm or distress, thereby disrupting the child's physical or psychological health and development." This refined definition highlighted the impact of adversity as being dependent on the intensity and frequency of exposure to adversities, which was not accounted for in the original definition (Felitti et al., 1998). It also pinpointed that adversities can occur in one's family/home and the larger society/community that they live in. These results also corroborate the findings presented by

**Table 4.** Composition of articles in the clusters of document co-citation analysis

| Cluster | Theme | Important articles |
| --- | --- | --- |
| Cluster 1 (red lines in Figure 3) | Concept and Assessment of ACEs | Dong et al., 2004; Finkelhor et al., 2013; Kalmakis and Chandler, 2014; Cronholm et al., 2015; Merrick et al., 2017; Mersky et al., 2017; Finkelhor, 2018 |
| Cluster 2 (green lines in Figure 3) | ACEs and Health Implications | Anda et al., 2006; Brown et al., 2009; Gilbert et al., 2015; Kalmakis and Chandler, 2015; Balistreri and Alvira-Hammond, 2016; Campbell et al., 2016; Hughes et al., 2017; Petruccelli et al., 2019 |
| Cluster 3 (blue lines in Figure 3) | ACEs and Mental Health | Chapman et al., 2004; Dong et al., 2004; Anda et al., 2010; Nurius et al., 2015; Balistreri and Alvira-Hammond, 2016; Hunt et al., 2017; Merrick et al., 2017 |
| Cluster 4 (yellow lines in Figure 3) | ACEs and Juvenile Delinquency | Duke et al., 2010; Fox et al., 2015; Baglivio and Epps, 2016; Metzler et al., 2017; Fagan and Novak, 2018; Baglivio et al., 2020 |

Dong et al. (2004), which were critical in establishing the interrelatedness of adversities – a chain reaction of sorts. Research (Mersky et al., 2017) continued to explore the wide array of what constitutes childhood adversity and has identified family financial problems, food insecurity, homelessness and sibling death to also cause significant trauma.

Studies in this cluster also identified the importance and potential pitfalls in the assessment of ACE. For instance, initiating widespread screening for ACE in healthcare settings ought to be approached cautiously until critical questions regarding the effective implementation of interventions, outcomes and costs associated with ACE screening and defining specific aspects to screen for are addressed (Finkelhor, 2018).

### Cluster 2: ACE and Health Implications

Studies in this cluster investigated the physical health outcomes of exposure to childhood adversity. Three systematic reviews and two meta-analyses have been included in this cluster. ACE have been linked to higher premature death (Brown et al., 2009), risky behaviors and morbidity measures, such as obesity, diabetes, myocardial infarction, coronary heart disease, stroke and asthma (Gilbert et al., 2015; Campbell et al., 2016). High exposure to ACE has also been associated with developmental disruption and increased healthcare utilization (Kalmakis and Chandler, 2015), as well as memory deterioration and substance abuse issues (Anda et al., 2006), which could contribute to the intergenerational transmission of ACE (Hughes et al., 2017; Petruccelli et al., 2019). Balistreri and Alvira-Hammond (2016) reported low levels of physical well-being among adolescents with high ACE scores. The study also highlighted the moderating effect of family functioning in the relationship between exposure to ACE and adolescent well-being.

### Cluster 3: ACE and Mental Health

This cluster focused on the study of ACE, their mental health consequences, how it is perceived and responded to, and ways to mitigate the mental and emotional effects of ACE. Findings from Nurius et al. (2015) corroborate with earlier studies that exposure to early adversity carries a unique capacity to impair adult psychological well-being. Dong et al. (2004) underscored the interconnected nature of adversities in one's life, implying that exposure to one form of adversity can trigger a cascade or "chain reaction" leading to further adversities. This understanding is crucial in recognizing ACEs' complex and cumulative effects on individuals' health and well-being.

More specifically, the papers in this cluster investigated the impact of childhood emotional abuse and other adverse events on the development of depressive disorders during adulthood. The results proved a strong association between emotional abuse and neglect and the risk for lifetime depressive disorders up to decades after their occurrence in both men and women (Chapman et al., 2004). Further, exposure to three or more ACE has been shown to increase the likelihood of moderate to heavy drinking, drug use and suicide attempts during adulthood (Merrick et al., 2017). Children as young as 9 years old begin to show signs of behavioral problems after exposure to adversities, and this association varies with race/ethnicity, gender and level of maternal education (Hunt et al., 2017). Finally, the papers in this cluster highlighted the importance of comprehensively addressing and mitigating ACE to break the cycle of adversity, promote resilience, support community mental health initiatives and identify protective factors (Dong et al., 2004).

### Cluster 4: ACE and Juvenile Delinquency

Cluster 4 contained articles about experiencing childhood adversity and their intersection with various socioeconomic variables, including poverty, gender, employment status, level of education, neighborhood and race/ethnicity, which, according to Metzler et al. (2017), signifies overall life opportunities and access to them. Compared with those with no ACEs, individuals with higher ACE scores were more inclined to report dropping out of high school, being unemployed and residing in households below the poverty level. This evidence highlighted the potential impact of preventing early adversity on health and life opportunities, with ramifications extending across generations.

However, a majority of the papers in this cluster dealt with the association between early adversity, violent behaviors and juvenile delinquency (Baglivio and Epps, 2016). Each form of ACE was notably linked with engaging in interpersonal violence during adolescence, encompassing various behaviors, such as delinquency, bullying, physical altercations, dating violence and carrying weapons on school premises, as well as self-directed violence, such as self-harm, suicidal thoughts and suicide attempts, with each additional adversity reported by youth, the likelihood of perpetrating violence increased by 35%–144% (Duke et al., 2010). Childhood maltreatment directly affected offending trajectories (Baglivio and Epps, 2016), and studies have particularly investigated racial differences in the experience of childhood adversity. Results from a study conducted by Fagan and Novak (2018) showed statistically significant differences among races for alcohol consumption, marijuana usage and arrest rates particularly among Blacks, with no similar impact observed among white counterparts. These findings have implications for policy change in the juvenile justice system, that is, recognizing and considering trauma histories of youth offenders when determining the legal course of action as critical to the rehabilitation and reintegration of children who offend (Fagan and Novak, 2018). The studies in this cluster offer insight into systemic and policy-level implications in mitigating childhood adversity.

### Discussion

The findings of this review offer several critical implications for researchers interested in ACE. The increasing trend in research focusing on ACE reflects an increasing acknowledgment of early-life events' profound impact on one's long-term health and well-being. However, the analysis revealed a noticeable rise in ACE-related publications between 2019 and 2023, coinciding with the global COVID-19 pandemic. It is important to note that while Figure 2 includes publication data from 2004 to March 2024, the trendline analysis has been restricted to the years 2004–2023. This decision was made to avoid skewing the trend due to incomplete data for 2024, which may not accurately reflect annual research output. While the study did not explicitly aim to assess the pandemic's influence, the observed trend aligned with the broader societal and academic recognition of the pandemic's impact on childhood adversity. The pandemic significantly exacerbated risk factors associated with ACE, such as social isolation, economic instability, disruptions to education and healthcare and increased cases of intimate partner violence (Taub, 2020; Anderson et al., 2022). These challenges definitely rendered children more vulnerable to physical, emotional and psychological harm (Imran et al., 2020), potentially driving scholarly interest in ACEs as a critical topic of investigation during this period. The increase in

ACE-related research during this time period also reflects the increased focus on resilience and coping mechanisms in navigating adversity (Sanders, 2020). This aligned with the study's broader aim of identifying trends and emerging themes in ACE literature. However, further analysis would be required to understand whether the rise in publications during this period was a continuation of an existing upward trend or if it was primarily driven by the pandemic.

While the increasing scholarly interest in ACE is evident, an important observation from this analysis was the significant disparity in research contributions across different regions of the world. The findings revealed that the field is predominantly shaped by researchers in Western countries, particularly in the United States, the United Kingdom, Canada and Australia. These nations have consistently led in publication volume and citation impact, underscoring their central role in driving knowledge production in this domain. Among the developing nations, only China and South Africa have made a notable presence, leaving significant gaps in contributions from other regions, especially from the Global South. This Western dominance highlights the need to decolonize knowledge production in the child protection and ACE research sectors. Decolonizing ACE research involves diversifying the intellectual leadership, research priorities and methodological approaches to better represent the lived experiences and sociocultural realities of children in non-Western contexts. This would ensure that ACE-related policies and interventions are culturally relevant and inclusive.

In addition to regional disparities, the analysis highlighted key contributors who have significantly shaped the ACE research landscape. Journals from the field of psychology that focus on ACE and child/adolescent development, such as the *Journal of Interpersonal Violence Children and Youth Services Review*, *Psychological Trauma: Theory, Research, Practice and Policy* and *Frontiers in Psychology*, stand out as the most productive and impactful in publishing ACE-related research. These journals have consistently disseminated key studies that have shaped both theoretical and practical understandings of ACEs. Among the most prolific researchers in this field are Kevin T. Wolff (John Jay Research and Evaluation Center), Melissa S. Jones (Brigham Young University), Hayley Pierce (Brigham Young University), Trevor Spratt (Trinity College Dublin) and Elizabeth Crouch (University of South Carolina). Notably, Kevin T. Wolff and Melissa T. Merrick (Prevent Child Abuse America) emerge as the most impactful authors, as reflected in their high citation rates. Their influential work on the role of neighborhood context and community in exposure to ACE, particularly among juvenile offenders, as well as their studies on the factors contributing to juvenile recidivism, have garnered significant academic attention and contributed to key discussions in the field of childhood adversity.

The results of the cluster analysis provided important insights into the evolving landscape of ACE research and highlighted the growing complexity of the field. The identification of four distinct clusters – each focusing on different aspects of ACE – demonstrated how the domain has expanded beyond initial conceptions of childhood adversity as solely an individual health issue.

Cluster 1, which focused on the definition and assessment of ACE, reflected ongoing efforts to refine and broaden the understanding of adversity, with an emphasis on the need for more inclusive representation across different populations. This shift is crucial in ensuring that ACE research accounts for the diverse experiences of individuals from various socioeconomic, racial and cultural backgrounds. The call for careful consideration of ACE

screening in healthcare settings underscored the importance of balancing the widespread adoption of these tools with ethical concerns, particularly regarding the inclusion of ACE in healthcare interventions. This theme aligned with the broader movement in the field toward reconceptualizing ACE, moving away from a purely individualistic perspective to one that includes social and environmental influences. The focus of Cluster 2 on the physical health outcomes of ACEs expanded the scope of ACE research by reinforcing the long-term implications of childhood adversity on adult health. The significant links between ACE and a range of chronic health conditions underscore the need for long-term public health strategies that address not only the immediate effects of adversity but also its intergenerational transmission. This cluster's findings provided critical support for integrating ACE-informed policies into healthcare practices. The mental health impact of ACE, explored in Cluster 3, continues to be a central concern in the field. The relationship between ACEs and mental health issues, such as depression, substance abuse and suicidal behaviors, highlighted the enduring psychological toll of early adversities. This cluster emphasized the need for comprehensive, trauma-informed approaches to mental health care that address not only the symptoms of mental health disorders but also the root causes, including childhood trauma. Furthermore, the focus on resilience and community mental health initiatives in this cluster provided a hopeful perspective, suggesting that addressing ACE through preventive and therapeutic means can break the cycle of adversity and promote healing. Finally, Cluster 4, which examined the link between ACE and juvenile delinquency, underscored the importance of understanding how early adversity influences social outcomes, such as criminal behavior and recidivism. This cluster highlighted the potential for early intervention and social support in mitigating the effects of ACE on one's life trajectories. The recognition of the lasting societal impacts of ACE, particularly in relation to education, poverty and access to resources, calls for a more systemic approach to ACE research and intervention. These findings further support the argument for addressing ACE not only at the individual level but also through policies that address the broader social determinants of health and well-being.

While the co-citation analysis identified four distinct thematic clusters, these clusters are not mutually exclusive; overlaps exist between them, reflecting the interconnected nature of ACE research. Overall, the cluster analysis revealed a comprehensive and evolving intellectual structure within ACE research. While the well-documented associations between ACE and health outcomes (Clusters 2 and 3) continue to be foundational to the field, this analysis offered a unique contribution by highlighting the intersection and evolution of these research strands. The identification of a distinct "ACE and Society" cluster (Cluster 4) marked a shift toward a more holistic understanding of ACE, acknowledging their societal implications beyond individual health outcomes. This broader conceptualization aligned with growing calls in the literature for a more integrated approach that addresses both individual and systemic factors in ACE research and intervention.

## Limitations

This study has several limitations that should be considered when interpreting its findings. First, the analysis was limited to the Scopus database, which, while comprehensive, may not capture all relevant publications and may have potentially excluded valuable insights from non-Scopus-indexed journals. Second, while citation metrics are a commonly used measure of influence, they do not always

capture the full impact of research, such as its practical application, policy influence or societal impact, which may limit the understanding of an author's or journal's true influence. Additionally, while the co-citation analysis revealed four thematic clusters, there may be overlaps or interconnections between these clusters that were not fully explored, which may have oversimplified the complex interrelations between the thematic areas. Lastly, this study focused specifically on co-citation patterns to map the intellectual structure of ACE research. Other bibliometric tools, such as keyword co-occurrence analysis, a valuable bibliometric technique for identifying thematic trends and their evolution over time, could be incorporated into future studies to offer deeper insights into thematic developments within the field.

### Future research

Given the dominance of research on ACE in developed countries, it is imperative to expand ACE research in developing nations, particularly in the Global South. Research conducted in these regions can bring unique insights, informed by local sociocultural realities, and help develop context-specific approaches to mitigate the effects of ACE. Strengthening research capacity in underrepresented regions, promoting global collaborations and increasing funding for non-Western researchers will be essential for diversifying knowledge production and addressing regional gaps. Additionally, revisiting Western-centric frameworks and promoting decolonization efforts in ACE research will help ensure that the field's growth reflects the diversity of children's experiences worldwide. Future research should also prioritize continued longitudinal studies to deepen our understanding of how ACE manifest over time and intersect with other social variables, such as caste, religion, gender identity and sexual orientation. This can aid in the development of targeted interventions to address the needs of diverse and vulnerable populations.

Moreover, future studies could examine the impact of global crises, such as the COVID-19 pandemic, on ACE research. A comparison of pre-pandemic and pandemic-era publications could provide clearer insights into how such crises influence scholarly priorities and the evolution of ACE research. Finally, identifying resilience factors and protective mechanisms against the adverse effects of ACE should be a key focus. Future research should explore individual, familial and community-level interventions that can promote resilience and mitigate the long-term impacts of adversity.

### Implications for practice

The findings of this bibliometric analysis highlight the critical need for trauma-informed practices across systems that interact with children. Educators, healthcare providers and social workers should integrate trauma awareness into their daily routines to create supportive environments that mitigate the long-term impacts of ACE. Preventative interventions targeting vulnerable populations, such as early childhood programs and community-based initiatives, can play a pivotal role in reducing health risks, mental health challenges and juvenile delinquency associated with ACE.

This study also emphasizes the need for localized research to inform culturally appropriate policies, particularly in underrepresented regions, such as the Global South. Policymakers should prioritize funding for research and interventions that address systemic inequalities and contextualize ACE prevention strategies. Additionally, professional training programs must equip healthcare and social service workers with the skills to identify and address ACE within diverse sociocultural contexts.

Global collaboration is essential for bridging research and practice gaps. International partnerships between nations, institutions and practitioners can enhance the global understanding of ACE and support the development of scalable and equitable interventions. By applying these insights, professionals will be able to help break the cycle of adversity and promote resilience in communities worldwide.

### Conclusion

In conclusion, this study highlighted the growing interest and recognition of ACE as a critical area of research, particularly in the context of the COVID-19 pandemic, which exacerbated risk factors associated with childhood adversity. The surge in ACE-related publications during this period not only reflects an intensified scholarly response to the pandemic's psychosocial impacts but also underscores the need to integrate ACE surveillance into future public health preparedness frameworks. The findings revealed a clear dominance of Western countries in shaping the discourse on ACE, with key journals and influential authors driving the field's development. The co-citation analysis identified four major thematic clusters – Concept and Assessment of ACE, ACE and Health Implications, ACE and Mental Health, and ACE and Society – which together provide a comprehensive overview of the intellectual structure of ACE research. This study offers a novel contribution by mapping the evolution of ACE research and revealing key themes, influential authors, journals and geographic disparities. Notably, the recognition of ACE as a societal issue, extending beyond individual health outcomes, marks an important shift toward more holistic approaches to addressing childhood adversity. These findings provide valuable insights for both researchers and policymakers, emphasizing the need for continued global collaborations to address the multifaceted impacts of ACE on individuals and society as a whole.

**Open peer review.** To view the open peer review materials for this article, please visit http://doi.org/10.1017/gmh.2025.10009.

**Data availability statement.** Data availability is not applicable to this article as no new data were created or analyzed in this study.

**Author contribution.** Conceptualization: B.K.S.S., G.K.J.; Data curation: B.K.S.S., G.K.J.; Formal analysis: B.K.S.S.; Software: B.K.S.S., G.K.J.; Supervision: G.K.J.; Writing – original draft: B.K.S.S.; writing – review and editing: B.K.S.S., G.K.J.

**Financial support.** This research received no specific grant from any funding agency, commercial or not-for-profit sectors.

**Competing interests.** The authors have no competing interests.

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
