## [Reviewer Report]

The study is well written but the figures need improvement. Please consider making the following corrections-

1. Title- Mention duration of included studies

2. Abstract- Please consider mentioning the name of the software tool used.

3. Introduction- well written.

4. Methodology- well written but a flow chart of study selection process will bring more clarity and transparency in reporting. Also, the following line needs to be rephrased-

a. 1st line- “The bibliometric methodology employs quantitative techniques, such as publication and citation analysis, to analyze units of publication (Broadus, 1987). ”....not appropriate. The meaning is not clear.

5. Results- Well reported.

6. Figures- Figure 2- Please consider increasing the font size and consider mentioning the thresholds in the figure legends.

Figure 3- Missing

7. Discussion- Limitations should be part of discussion section, followed by future research recommendations

I hope the comments would be helpful in improving the quality of the manuscript.

Best Wishes

---

## [Reviewer Report]

The topic addressed in this work is highly relevant, as the authors employ bibliometric methods—well-established techniques that are effectively used to track and analyze research status, trends, gaps, challenges, and priorities across various scientific fields. These methods are invaluable for policymakers and researchers in identifying research dynamics, emerging frontiers, future directions, and investment opportunities. The analysis focuses on the scientific literature related to adverse childhood experiences (ACEs). However, there are several key issues that need to be addressed in the revised version:

1. In the abstract, you mention that the leading journal is “Children and Youth Services Review.” However, upon reviewing Table 2, it is clear that “Journal of Interpersonal Violence” is the most productive journal, while “Children and Youth Services Review” has the highest number of citations. Please clarify this discrepancy in the abstract by highlighting the most productive journal.

2. In the abstract, when referring to the most productive countries, journals, and authors, it would be helpful to include their productivity along with percentages (e.g., n, %).

3. The timeframe of the analysis should be revised. You state that it extends from 2004 to 2024, but the data was collected through March 2024. This means that the full year of 2024 has not been covered, and the timeframe should be accurately reflected.

4. A strong concluding statement should be added at the end of the abstract.

5. On Page 2, Lines 36-55, when citing the work of “Felitti et al.”, revise the citation style (e.g., Cronholm et al., 2015) to: the Philadelphia Expanded ACE Study by Cronholm et al. (2015).

6. Please ensure that the citation style throughout the text is consistent with the journal’s preferred format.

7. In the Methods section, I recommend including examples of bibliometric studies conducted in various fields of science. For example, you may reference recent bibliometric studies on One Health, Climate change, SDGs, etc.

8. In the data extraction and cleanup section, clearly define whether the search was conducted on titles, titles and abstracts, or titles, abstracts, and keywords.

9. It is recommended to include a figure showing data inclusion and exclusion, following PRISMA guidelines. This will make the methodology of identification, screening, inclusion, and exclusion clearer to readers.

10. The number of publications analyzed should be included in the abstract section.

11. In the data analysis section (Lines 26-28), please revise the paragraph for clarity and correct grammatical errors. For example, “The analysis was carried out using Microsoft Excel 2019, and VOSviewer software (version 1.6.20) was used as a bibliometric analysis and visualization tool (Eck et al., 2010).”

12. Since you used VOSviewer software, why did you not conduct a keyword co-occurrence analysis? This analysis is effective for revealing major topics and themes, as well as their evolution over time, through overlay visualization maps.

13. In Figure 1, add titles for both the x-axis and y-axis. Additionally, it would be beneficial to include the equation of the trend (e.g., linear, exponential) and the coefficient of determination (R²).

14. In Table 2, I suggest incorporating the impact factors of the listed journals.

15. In Table 3, include the affiliations of the most productive authors.

16. The impact of COVID-19 on research directions was not discussed, although you mentioned in the abstract that “A notable finding was the surge in ACE-related publications during the COVID-19 pandemic, potentially reflecting increased global attention on childhood adversity amid heightened social and economic challenges.” This issue should be addressed in the discussion to strengthen your conclusion.

17. I advise the authors to extend the work by conducting a keyword co-occurrence analysis.

---

## [Reviewer Report]

One concern regarding Figure 2 is the inclusion of publication data from the year 2024 in establishing the publication trend (exponential trend) and calculating the coefficient of determination (R²). Since the year 2024 is not yet complete, including its data in the trend analysis may lead to inaccurate results and a misleading representation of the overall trend. It is recommended to retain the productivity data for 2024 in the figure; however, when establishing the trend line and its corresponding equation, the data for 2024 should be excluded. Additionally, this should be clearly stated and explained in the figure caption or the discussion.